# Atmospheric impacts of chlorinated very short-lived substances over the recent past - Part 2: Impacts on ozone

Ewa M. Bednarz[1,2,3], Ryan Hossaini[1,4], Martyn P. Chipperfield[5,6]

1. Lancaster Environment Centre, Lancaster University, Lancaster, UK
2. Cooperative Institute for Research in Environmental Science (CIRES), University of Colorado Boulder, Boulder, CO, USA.
3. NOAA Chemical Sciences Laboratory (NOAA CSL), Boulder, CO, USA
4. Centre of Excellence in Environmental Data Science (CEEDS), Lancaster University, Lancaster UK.
5. School of Earth and Environment, University of Leeds, Leeds, UK
6. National Centre for Earth Observation (NCEO), University of Leeds, Leeds, UK

Correspondence to: Ewa M. Bednarz (ewa.bednarz@noaa.gov)

**Abstract**

Depletion of the stratospheric ozone layer remains an ongoing environmental issue, with increasing stratospheric chlorine from Very Short-Lived Substances (VSLS) recently emerging as a potential but uncertain threat to its future recovery. Here the impact of chlorinated VSLS on past ozone is quantified, for the first time, using the UM-UKCA chemistry-climate model. Model simulations nudged to reanalysis fields show that in the second decade of the 21st century Cl-VSLS reduced total column ozone by, on average, ~2-3 DU in the springtime high latitudes and by ~0.5 DU in the annual mean in the tropics. The largest ozone reductions were simulated in the Arctic in the springs of 2011 and 2020. During the recent cold Arctic winter of 2019/2020 Cl-VSLS resulted in local ozone reductions of up to ~7% in the lower stratospheric and ~7 DU in total column by the end of March.

Despite ~doubling of Cl-VSLS contribution to stratospheric chlorine over the early 21st century, the inclusion of Cl-VSLS in the nudged simulations does not substantially modify the magnitude of the simulated recent ozone trends and, thus, do not help to explain the persistent negative ozone trends that have been observed in the extra-polar lower stratosphere. The free-running simulations, on the other hand, suggest Cl-VSLS induced amplification of the negative tropical lower stratospheric ozone trend by ~20 %, suggesting a potential role of the dynamical feedback from Cl-VSLS induced chemical ozone loss. Finally, we calculate the ozone depletion potential of dichloromethane, the most abundant Cl-VSLS, at 0.0107. Our results illustrate a so-far modest but nonetheless non-negligible role of Cl-VSLS in contributing to stratospheric ozone budget over the recent past that if to continue could offset some of the gains achieved by the Montreal Protocol.

## 1. Introduction

Depletion of the stratospheric ozone layer remains an ongoing environmental issue, caused predominantly by long-lived ozone-depleting substances (ODSs) containing chlorine and bromine. Controls on the production of ODSs, such as chlorofluorocarbons, introduced by the Montreal Protocol and its amendments have successfully reduced the stratospheric loading of chlorine and bromine (e.g. Bernath and Fernando, 2018) and thus it is expected that ozone should return to pre-1980 levels in the middle to latter half of this century (WMO, 2022). However, in recent years it has been became evident that so-called Very Short-Lived Substances (VSLS), with lifetimes in the near-surface atmosphere of less than ~6 months, also provide a significant source of stratospheric halogens (e.g. Fernandez et al., 2014; Wales et al., 2018; Keber et al., 2020). While brominated VSLS (e.g. $CHBr_3$) are typically of natural ocean origin, recent studies have raised concerns that unregulated industrial emissions of chlorinated VSLS (Cl-VSLS) are offsetting some of the gains of the Montreal Protocol (e.g. Hossaini et al., 2019; Bednarz et al., 2022) and could thus delay future recovery of the ozone layer (Hossaini et al., 2017).

Dichloromethane ($CH_2Cl_2$) is a common solvent used in a wide variety of applications and is the most abundant atmospheric Cl-VSLS. Global $CH_2Cl_2$ emissions in 2020 were estimated at ~1.1 Tg/yr, a factor 2.5 increase from the year 2000 (WMO, 2022) that has been due predominately to growth in Asia (Claxton et al., 2020; An et al., 2021). The ozone depletion potential (ODP) of $CH_2Cl_2$ has been estimated to be ~0.01-0.02 (Claxton et al., 2019), though despite recent strong interest in this gas there has not been more estimates of this important policy metric. Other Cl-VSLS with significant industrial sources include chloroform ($CHCl_3$), Asian emissions of which have also grown substantially (Fang et al., 2019), 1,2-dichloroethane ($CH_2ClCH_2Cl$) and perchloroethylene ($C_2Cl_4$). In Part 1 of this study (Bednarz et al., 2022), we investigated the impacts of these Cl-VSLS on the stratospheric chlorine budget using the Unified Model coupled to the United Kingdom Chemistry and Aerosol (UM-UKCA, Walters et al., 2019; Archibald et al., 2020) chemistry-climate model (CCM). We showed that the contribution from these Cl-VSLS to stratospheric chlorine had increased from 70 ppt Cl in 2000 to 130 ppt Cl in 2019, i.e. almost doubling over the first two decades of the 21st century.

Evidence of ozone layer recovery is apparent in the polar stratosphere from observations and models (e.g. Solomon et al., 2016; Kuttippurath et al., 2018; WMO, 2022). However, a persistent downward trend in extra-polar lower stratospheric ozone has been reported from datasets based on satellite observations (e.g. Ball et al., 2018; 2019). In this region, ozone is strongly affected by dynamical variability (Chipperfield et al., 2018) and the downward ozone trend is likely associated with large-scale changes to atmospheric circulation (Wargan et al., 2018; Orbe et al., 2020) or its variability (Stone et al., 2018). While the effect of Cl-VSLS on the tropical lower stratospheric ozone trend in a chemistry-transport model has been estimated to be small (Chipperfield et al., 2018), a larger impact has recently been reported using a global chemistry-climate model containing a coupled troposphere-stratosphere chemistry scheme including chlorine, bromine and iodine VSLS (Villmayor et al., 2023),

and as such the issue should still be re-examined. Moreover, the effects of Cl-VSLS on ozone more broadly, including their contribution to some of the strong Arctic ozone depletions observed in the recent past (e.g. Feng et al., 2021), is unknown.

The impacts of Cl-VSLS on stratospheric ozone and its trends are thus the focus of this Part 2 of our study. Part 1 (Bednarz et al., 2022a) highlighted important differences in the stratospheric Cl-VSLS levels simulated in free-running and nudged UM-UKCA model versions (including differences brought about by the choice of reanalysis used for nudging). Hence ozone impacts are investigated here using three sets of transient simulations over the recent past (1990 onwards), both with and

70 without Cl-VSLS included. These are: (1) VSLS and BASE that have free-running meteorology, (2) $VSLS_{SD-5}$ and $BASE_{SD-5}$ that are nudged to the ECMWF ERA5 reanalysis, and (3) $VSLS_{SD-I}$ and $BASE_{SD-I}$ that are nudged to the ECMWF ERA-Interim reanalysis. These simulations are described in more detail in the Methods section (Section M1). We quantify the impacts of Cl-VSLS on ozone over the beginning of the 21$^{st}$ century (Section 2), including the contribution of Cl-VSLS to the elevated ClO and reduced ozone observed during the recent very cold Arctic winter of 2019/2020 (Section 3). We also discuss the

75 contribution of Cl-VSLS to the recent ozone trends (Section 4), as well as use additional UM-UKCA simulations (Section M2 in Methods) to calculate ODP of $CH_2Cl_2$ (Section 5). Summary and conclusions are given in Section 6.

## 2. Impacts on ozone in the second decade of 21$^{st}$ century

Figure 1 shows the difference in total column ozone between the integrations with and without Cl-VSLS as a function of latitude and time (from January 2010 onwards), for the simulations nudged to either ERA5 (Fig.1a) or ERA-Interim (Fig. 1b)

reanalysis. The integrations nudged to both reanalysis datasets show springtime ozone losses of 2-3 DU on average in the Northern Hemisphere (NH) and Southern Hemisphere (SH) high latitudes during the second decade of the 21$^{st}$ century (Fig. 1c). When the simulations are nudged to ERA-5, the largest ozone reductions are simulated over the Arctic in the springs of 2011 and 2020 (7 DU and 5 DU zonal mean ozone loss, respectively, from Cl-VSLS in April monthly mean). These larger ozone losses were facilitated by the formation of particularly strong, cold and long-lasting polar vortex (Manney et al., 2011;

2022; Sinnhuber et al., 2011).

We note that while very similar average large scale ozone losses are diagnosed from the simulations nudged to different reanalysis products (Fig. 1c), some differences can emerge for individual regions and seasons. In particular, no significant Cl-VSLS-induced Arctic ozone loss is diagnosed for the spring 2011 from the simulations nudged to ERA-Interim, while the

90 Arctic ozone loss modelled in the spring of 2014 is notably higher in those run than in the runs nudged to ERA5. This might be related to the generally small and variable size and structure of the NH polar vortex, thus difficulties in reproducing its dynamical properties in a nudged model set up, or to the differences in the resolved transport between the two reanalyses (e.g. Diallo et al., 2021; Ploeger et al., 2021; Bednarz et al., 2022). These results thus suggest that the choice of reanalysis for nudging could also be important in some years for the diagnosed ozone impacts from Cl-VSLS.

In the tropics, Cl-VSLS reduce total column ozone by ~0.5 DU on average in the second decade of the 21$^{st}$ century (Fig. 1c), but the decreases can temporarily reach 1-2 DU is some years (Fig. 1b-c). Whilst small in absolute terms, these tropical ozone reductions can play a comparatively larger role for surface UV due to climatological ozone being much lower there than at higher latitudes, and due to the smaller daytime solar zenith angles.

The corresponding vertically resolved ozone changes are shown in Fig. 2. The inclusion of Cl-VSLS results in ~0.5-1 % yearly mean ozone reductions in the tropical lower and upper stratosphere on average over the second decade of the 21$^{st}$ century (Fig. 2a). Larger percentage ozone reductions of up to ~4-4.5% are found in the Antarctic lower stratosphere during spring (Fig. 2b). Overall qualitatively and quantitatively similar O$_3$ responses are found if only the last 3 years of the integrations (2017-2019) are considered (Fig. S1), i.e. when the contribution of Cl-VSLS to the stratospheric chlorine budget is largest (Bednarz et al., 2022).

Given the significant dynamical variability characterising ozone levels on year-to-year timescales, we focus in this section on the results from the nudged model simulations only. We note that whilst the corresponding free-running UM-UKCA simulations suggest higher Cl-VSLS induced lower stratospheric ozone losses (Fig. S2), consistent with the larger Cl-VSLS product gas to source gas stratospheric chlorine injection (Bednarz et al., 2022), there is large uncertainty in these values due to the contribution of natural variability.

## 3 Impacts during Arctic winter 2019/2020

The recent decade has seen a number of strong Arctic ozone depletion episodes reported from the observational record (WMO, 2018). Amongst these was the Arctic winter of 2019/2020, where the formation of strong, cold and relatively undisturbed polar vortex led to one of the largest Arctic ozone depletions observed in the recent past (e.g. Manney et al., 2020; Feng et al., 2021; Wohltmann et al., 2020; Lawrence et al., 2020; Inness et al., 2020; Grooß and Müller, 2021).

Consistent with the observations, significantly elevated ClO concentrations (up to 800 ppt ClO at 50 hPa on 1 March 2020, Fig. 3a) were simulated in the Arctic in spring in the UM-UKCA simulation nudged to the ERA5 reanalysis (VSLS$_{SD5}$). Comparison with the BASE$_{SD5}$ run that did not include Cl-VSLS shows differences up to ~25 ppt of ClO (Fig. 3d). Increased chlorine- and bromine-catalysed ozone depletion along with reduced transport of higher ozone levels from the mid-latitudes and/or higher altitudes resulted in very low ozone levels simulated in the Arctic at the end of March. Ozone levels of less than 1 ppb at 50 hPa were simulated in VSLS$_{SD5}$ on 31 March (Fig. 3b), corresponding to the minimum in the total column values of less than 240 DU at the same time (Fig. 3c). We find that Cl-VSLS on their own reduced ozone locally by up ~7 % at 50

hPa (Fig. 3e) and by up to ~7 DU in total column by the end of March (Fig. 3f). Similar total column ozone losses were found also in early April (Fig. S3)

In comparison, the impact of curbing emissions of long-lived ODSs achieved by the Montreal Protocol was estimated using the TOMCAT/SLIMCAT chemistry-transport model to reduce the magnitude of the Arctic ozone depletion in that spring by up to ~35 DU in mid-March compared to the peak halogen levels in early 2000 (Feng et al., 2021). This illustrates that Cl-VSLS emissions have played a modest but nonetheless important contribution to one of the largest stratospheric ozone depletion episodes observed in the Arctic, and by doing so acted to significantly offset some of the environmental gains achieved by the Montreal Protocol to date.

## 4. Contribution to the recent ozone trends

Despite ongoing recovery of stratospheric ozone, observational evidence suggests existence of negative ozone trends over the recent past in the tropical and mid- latitude lower stratosphere. The causes behind these are still not fully understood (Ball et al., 2018, 2020), although the contribution of dynamical changes and/or variability in atmospheric circulation is likely important (Wargan et al., 2018; Stone et al., 2018; Orbe et al., 2020). Both vn2.6 and vn2.7 of the SWOOSH merged satellite ozone product (Davis et al., 2016) show negative ozone trends over 2000-2019 throughout the tropical lower stratosphere and in the mid-latitudes of both hemispheres at the altitudes of ~150 hPa and ~50 hPa (Fig. 4a-b; see Section M3 in Methods for the details of the trend calculations).

All UM-UKCA simulations used here show negative ozone trends over the same period in the tropical lower stratosphere (Fig. 4c-h), in line with the GHG-induced acceleration of upwelling in the tropical troposphere (not shown). All ERA5 and ERA-Interim nudged simulations also reproduce qualitatively the observed negative trends in the SH mid-latitudes, with statistically significant trends that maximise at two altitudes in the lower stratosphere. This is not the case for the free-running simulations, which show positive trends in the SH instead and a suggestion of small negative (but statistically not significant) ozone trends in the NH mid-latitudes. This highlights that the UM-UKCA model is capable of reproducing some of the negative lower stratospheric ozone trends seen from the observations, but the exact structure of the response depends on the choice of model-set up, highlighting the importance of the model dynamical fields in reproducing the observed response.

Regarding the role of Cl-VSLS, we find that the magnitudes of the stratospheric ozone trends are very similar between the pairs of nudged simulations with and without Cl-VSLS, with only slightly more negative or less positive trends with the inclusion of Cl-VSLS (Figure 5 and S3). This suggest that the purely chemical impacts of increasing Cl-VSLS alone over the recent past are unlikely to be the main contributor to the negative lower stratospheric ozone trends reported from observations. This is thus in agreement with the conclusion of Chipperfield et al. (2018) that used a chemistry-transport model. Interestingly,

the Cl-VSLS induced modulation of lower stratospheric ozone trends is somewhat larger when inferred from the free-running UM-UKCA simulations. In the tropics (25°S-25°N), Cl-VSLS amplify the decrease in the tropical lower stratospheric ozone by around ~20% (i.e. from ~2.5% $O_3$ per decade in BASE to ~3% $O_3$ per decade in VSLS, Fig. 5). This result suggests a possible dynamical feedback from Cl-VSLS induced ozone loss on atmospheric circulation and ozone transport, which would not be represented in a nudged model configuration. Although longer simulations would be needed to confidently diagnose such impact, the finding broadly supports the recent study by Villamayor et al. (2023) who used a free-running configuration of the Community Earth System Model to demonstrate a contribution of VSLS to the tropical lower ozone trends in their model. Note, however, that their study considered both natural and anthropogenic VSLS, including long-term changes in bromine and iodine species (which were not included in our study), thus a direct quantitative comparison with our work is not possible.

## 5. Ozone Depleting Potential of $CH_2Cl_2$

In the final part of this study, we quantify the ODP and stratospheric ODP of $CH_2Cl_2$ using the time-slice simulations described in the Methods section (Section M4). ODP is an important and well-established metric that is reported in WMO/UNEP Ozone Assessment Reports and other policy-facing documents to gauge the possible ozone depleting effect of a gas relative to CFC-11. Unlike for long-lived species, there are few explicit (i.e. based on global model calculations) ODP estimates of VSLS in the literature. This in part reflects the relative complexity of a VSLS ODP calculation, which requires consideration of both the source gas and product gas injection of halogens to the stratosphere. A sensitivity of the ODP to emission location and season can also play a role for some species (e.g. Brioude et al., 2010). Given the significant upward trend in the $CH_2Cl_2$ production and emission from its predominantly industrial source, the quantification of ODP for $CH_2Cl_2$ is particularly important.

The responses of modelled annual mean ozone to the CFC11 and $CH_2Cl_2$ perturbations are shown in Fig. S5, and the global mean changes are summarised in Table 1. From these data, we calculate the $CH_2Cl_2$ ODP of 0.0107 (± 0.0064-0.0175, Table 1). This result constitutes, to our knowledge, only the second estimate of $CH_2Cl_2$ ODP in literature, and falls within the range of 0.0097-0.0208 reported in Claxton et al. (2019). The calculated stratospheric ODP of 0.0102 (±0.0062-0.0163) is similar to the whole atmosphere ODP metric, implying that $CH_2Cl_2$ has a relatively small effect on ozone below the tropopause in UM-UKCA. In part, this reflects the relatively long tropospheric lifetime of $CH_2Cl_2$ (~100 days in the boundary layer; Hossaini et al., 2019), especially compared to some particularly short-lived iodine species (e.g. $CF_3I$) for which the distinction between ODP and SODP can be particularly important (Zhang et al. 2020).

## 6. Summary and conclusions

By controlling the production and use of long-lived ozone-depleting substances, the Montreal Protocol has been immensely successful in reducing the abundance of atmospheric halogens (chlorine and bromine). In consequence, Earth's ozone layer is on a slow pathway to recovery. However, this landmark agreement faces new challenges, including the rapid growth of ozone-depleting chlorinated very short-lived substances which are not controlled by the Montreal Protocol or its amendments and adjustments. In this study, we have quantified for the first time the time-varying impact of uncontrolled Cl-VSLS emissions on stratospheric ozone, using the state-of-the-art UM-UKCA chemistry-climate model.

Model simulations nudged to reanalysis fields show that Cl-VSLS reduced total column ozone by, on average, ~2-3 DU in the springtime high latitudes and by ~0.5 DU in the annual mean over tropics in the second decade of the 21$^{st}$ century. In comparison, the ozone loss from the natural brominated VSLS emissions during the same time was estimated at ~1-2 DU in the tropics and ~5-6 DU in the midlatitudes (Barrera et al., 2020), albeit using a different climate model. Here, in the ERA5-nudged simulations, the largest ozone reductions were simulated in the Arctic in the springs of 2011 and 2020. We note some dependence of our Cl-VSLS results in specific regions and seasons on the choice of reanalysis used for nudging. We also quantified the Cl-VSLS impacts during the recent Arctic winter of 2019/2020, where the formation of a strong and cold polar vortex led to one of the largest Arctic stratospheric ozone depletion episodes in the observational record. In this case, Cl-VSLS resulted in up to ~7 % local reduction of lower stratospheric ozone by the end of March, contributing to ~7 DU local ozone depletion to the overall Arctic ozone anomaly.

Regarding recent ozone trends, the UM-UKCA model is shown to be capable of reproducing the negative lower stratospheric ozone trends reported from the satellite observations in the tropics and the SH mid-latitudes, but with the exact structure of the response depending on the choice of model-set up, indicating the importance of the model dynamical fields in reproducing the observed response. Importantly, the inclusion of Cl-VSLS does not substantially modify the magnitude of trends diagnosed from the nudged simulations. However, slightly larger effect is inferred from the free-running simulations, with Cl-VSLS amplifying the negative tropical lower stratospheric ozone trend by ~20 %, suggesting a potential role of the dynamical feedback from Cl-VSLS induced chemical ozone loss in contributing to the simulated lower stratospheric ozone trends.

Our results illustrate a so-far modest but nonetheless important role of Cl-VSLS in contributing to the stratospheric ozone budget over the recent past. If the growth in Cl-VSLS emissions inferred in the last decade (Feng et al., 2018; Fang et al., 2019, Claxton et al., 2020) is to continue into the future, these gases could exert a larger influence on future stratospheric ozone levels and, thus, continue to offset some of the gains achieved by the Montreal Protocol and delay the recovery of the ozone layer.

## Acknowledgements

The authors acknowledge support from the UK Natural Environment Research Council (NERC) SISLAC project (NE/R001782/1), NERC Independent Research Fellowship (NE/N014375/1), and NERC ISHOC project (NE/R004927/1). EMB also acknowledges support from the NOAA cooperative agreement NA22OAR4320151.

The simulations were carried out using MONSooN2, a collaborative high performance computing facility funded by the Met Office and the Natural Environment Research Council, and using the ARCHER UK National Supercomputing Service.

## Conflicting interests

The authors declare they have no conflict of interests.

**Methods**

**M1. Transient 1990-2019 UM-UKCA simulations**

We use vn11.0 of the UM-UKCA CCM (Walters et al., 2019; Archibald et al., 2020), run in atmospheric-only mode with prescribed observed sea-surface temperatures and sea-ice. The chemistry scheme used is the recently developed Double Extended Stratospheric-Tropospheric (DEST vn1.0; Bednarz et al., 2022b) scheme that includes comprehensive stratospheric halogen chemistry. The simulations analysed here are described fully in Bednarz et al. (2022a). Briefly, they consist of 3 pairs – with and without Cl-VSLS - of transient 1990-2019 (or 1990-2020) experiments. Simulations with Cl-VSLS used imposed

time- and latitudinally- varying lower boundary conditions (LBC), derived using surface Cl-VSLS measurements from NOAA and AGAGE stations. The first pair of runs, termed VSLS (i.e. with Cl-VSLS) and BASE (i.e. no Cl-VSLS), used a free-running meteorology, with 3 ensemble members each to reduce the contribution of natural variability. The second pair, $VSLS_{SD5}$ and $BASE_{SD5}$, used meteorology nudged to the ERA5 reanalysis (Hersbach et al., 2020). The third pair, $VSLS_{SDI}$ and $BASE_{SDI}$, used meteorology nudged to the ERA-Interim reanalysis (Dee et al., 2011).

**M2. Time-slice UM-UKCA simulations**

In addition to the transient simulations discussed above, we also performed a set of free-running 'time-slice' simulations under perpetual year 2015 conditions in order to calculate the ozone depletion potential (ODP) of $CH_2Cl_2$. In each case, the climatological sea-surface temperatures and sea-ice fields were the mean over the period 2011-2019 inclusive. Lower boundary conditions for ODSs and other long-lived gases for the year 2015 were taken from the SSP2-4.5 scenario, whilst the emissions

of other chemical tracers corresponded to the averages over 2015-2016 conditions. The meteorology in these runs is free running. The simulations include: a base run without Cl-VSLS; a simulation with an additional 100 ppt of CFC-11 at the surface relative to the 231 ppt CFC-11 in the base run; and a simulation with a 3 Tg/yr global $CH_2Cl_2$ emission flux (as opposed to the LBC used as a source of $CH_2Cl_2$ in the transient experiments described in Section M1 above). For the latter, emissions were assumed to be evenly distributed over North America, Europe, and South-East Asia (Fig. S6). All simulations were run

to steady-state, and then for additional 50 years that are used in the analysis.

**M3. Calculation of ozone trends**

In Section 4 we discuss linear trends in de-seasonalised ozone values from December 1999 to August 2019 for the ensemble mean free-running integrations as well as for each of the nudged runs. Following the procedure in Bednarz et al. (2022a), in each case zonal and monthly mean $O_3$ data are first interpolated onto a 10°-latitude grid and seasonally averaged (DJF, MAM,

JJA and SON). The resulting seasonal mean time series are then de-seasonalised (i.e. long-term mean for each season was removed), and a simple linear trend is calculated. The same procedure is also performed for calculating trends in the observed ozone values as given by the vn2.6 and vn2.7 of the SWOOSH merged satellite ozone dataset (Davis et al., 2016).

**M4. Calculation of CH$_2$Cl$_2$ ozone depletion potential**

The rate of CFC-11 emission corresponding to the 100 ppt surface increase is calculated at steady state, when the global emission of CFC-11 equates to its global loss (via photolysis and the reactions with O($^1$D) and OH). This is calculated to be 0.0350 Tg/yr, in good agreement with estimates reported in previous ODP studies (e.g. Wuebbles et al., 2011). The ODP of CH$_2$Cl$_2$ can then be calculated following Eq. (1), where ΔTCO3 denotes the global annual mean total column ozone change due to a unit emission of either CH$_2$Cl$_2$ or CFC-11:

ODP(CH$_2$Cl$_2$) = ΔTCO3(CH$_2$Cl$_2$) / ΔTCO3(CFC-11)                    (Eq. 1)

For VSLS that have a non-negligible impact on tropospheric ozone, 'stratospheric ODP' may provide a more informative metric where or when the goal is to evaluate the effect of a substance on the ozone layer (Zhang et al., 2020). In that case,
stratospheric ODP is calculated analogously using Eq. (2), where ΔSCO3 denotes the corresponding annual mean stratospheric column ozone change (here approximated by not including the airmasses at or below the tropopause):

Stratospheric ODP(CH$_2$Cl$_2$) = ΔSCO3(CH$_2$Cl$_2$) / ΔSCO3(CFC-11)                    (Eq. 2)

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

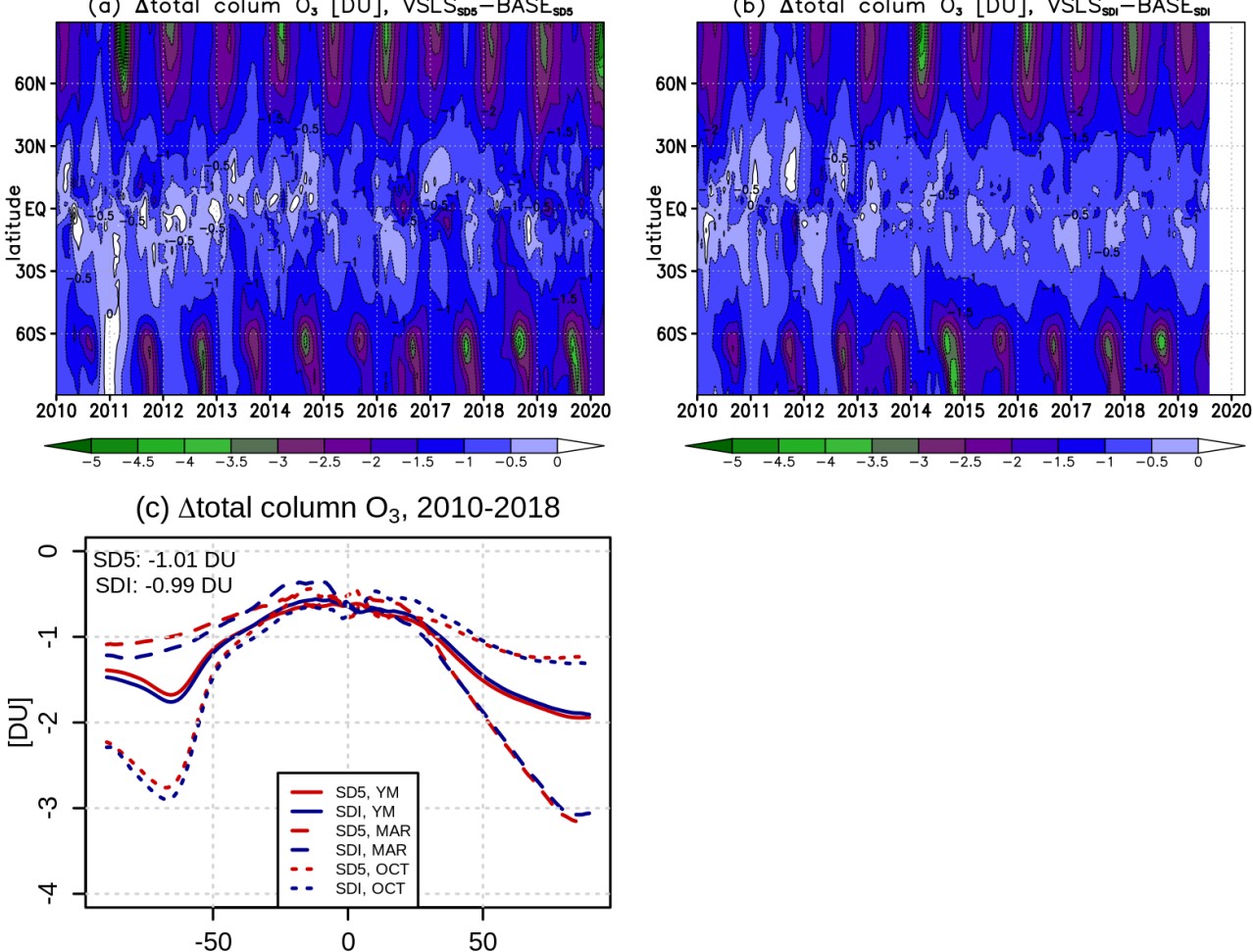

**Figure 1. The impacts of Cl-VSLS on recent total column ozone values.** Differences in monthly mean total column $O_3$ as a function of latitude and time (from January 2010 to April 2020) between the pairs of runs with and without Cl-VSLS nudged to either (a) ERA5 (VSLS$_{SD-5}$ and BASE$_{SD-5}$) or (b) ERA-Interim (VSLS$_{SD-I}$ and BASE$_{SD-I}$). Panel (c) shows the yearly mean (solid lines), March (dashed lines) and October (dotted lines) total column ozone differences averaged over 2010-2018 for runs nudged to ERA5 (red) and ERA-Interim (blue). The values shown in top left corner indicate the respective annual global mean total column ozone changes over that period.

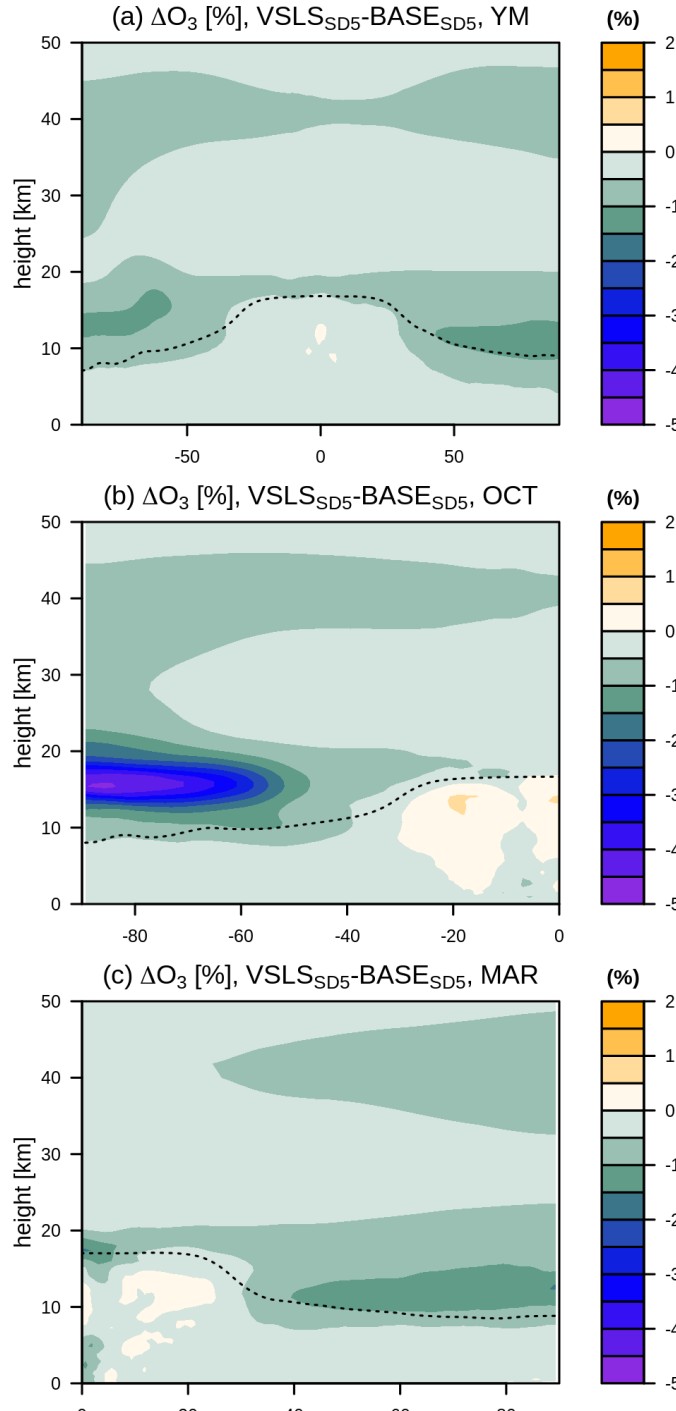

**Figure 2. The impacts of Cl-VSLS on recent stratospheric ozone levels.** Shading: Differences in 2010-2019 (a) yearly mean, (b) October and (c) March ozone [%] between the nudged VSLS$_{SD-5}$ and BASE$_{SD-5}$ runs. Dashed lines indicate the location of the model tropopause in VSLS$_{SD-5}$ for reference.

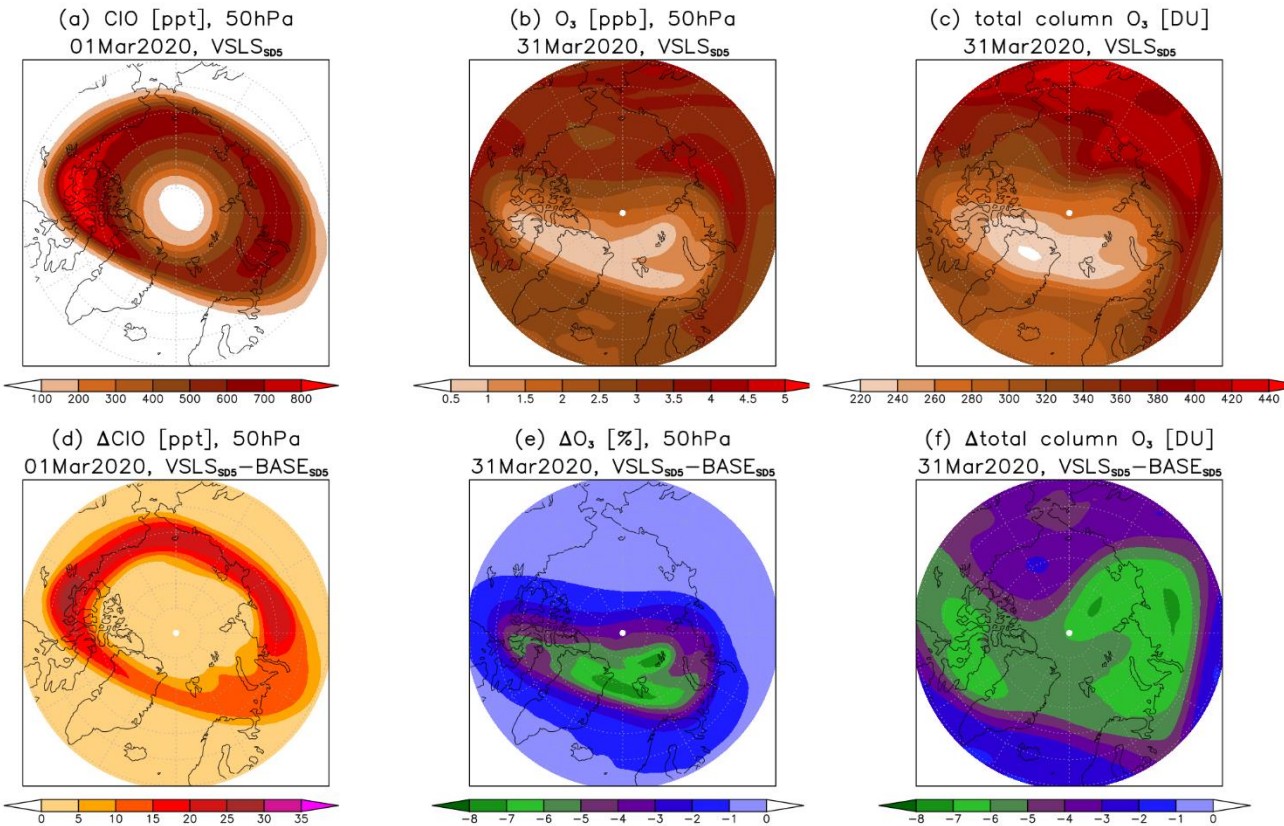

**Figure 3. The role of Cl-VSLS during Arctic winter of 2019/2020.** Stereographic projections poleward of 60°N of daily mean (a) ClO at 50 hPa [ppt] on 1 March 2020, (b) ozone at 50 hPa [ppb] on 31 March 2020 and (c) total column ozone [DU] on 31 March ] simulated in the nudged VSLS$_{SD-5}$ run. Shown in panels (d-f) are the respective differences between VSLS$_{SD-5}$ and BASE$_{SD-5}$ runs.

| Emissions | Δ(TCO3) (±2σ) | Δ(SCO3) (±2σ) |
|---|---|---|
| 3 Tg- $CH_2Cl_2$/yr | -3.06 DU (=1.0 %) ±0.80 DU | -2.76 DU ±0.72 DU |
| 0.0350 Tg-CFC11/yr | -3.34 DU (=1.1 %) ±0.76 DU | -3.17 DU ±0.68 DU |
| $CH_2Cl_2$ ODP | Total ODP | Stratospheric ODP |
| | 0.0107 (0.0064-0.0175) | 0.0102 (0.0062-0.0163) |

**Table 1. Summary of the terms in the calculation of $CH_2Cl_2$ ozone depletion potential.**

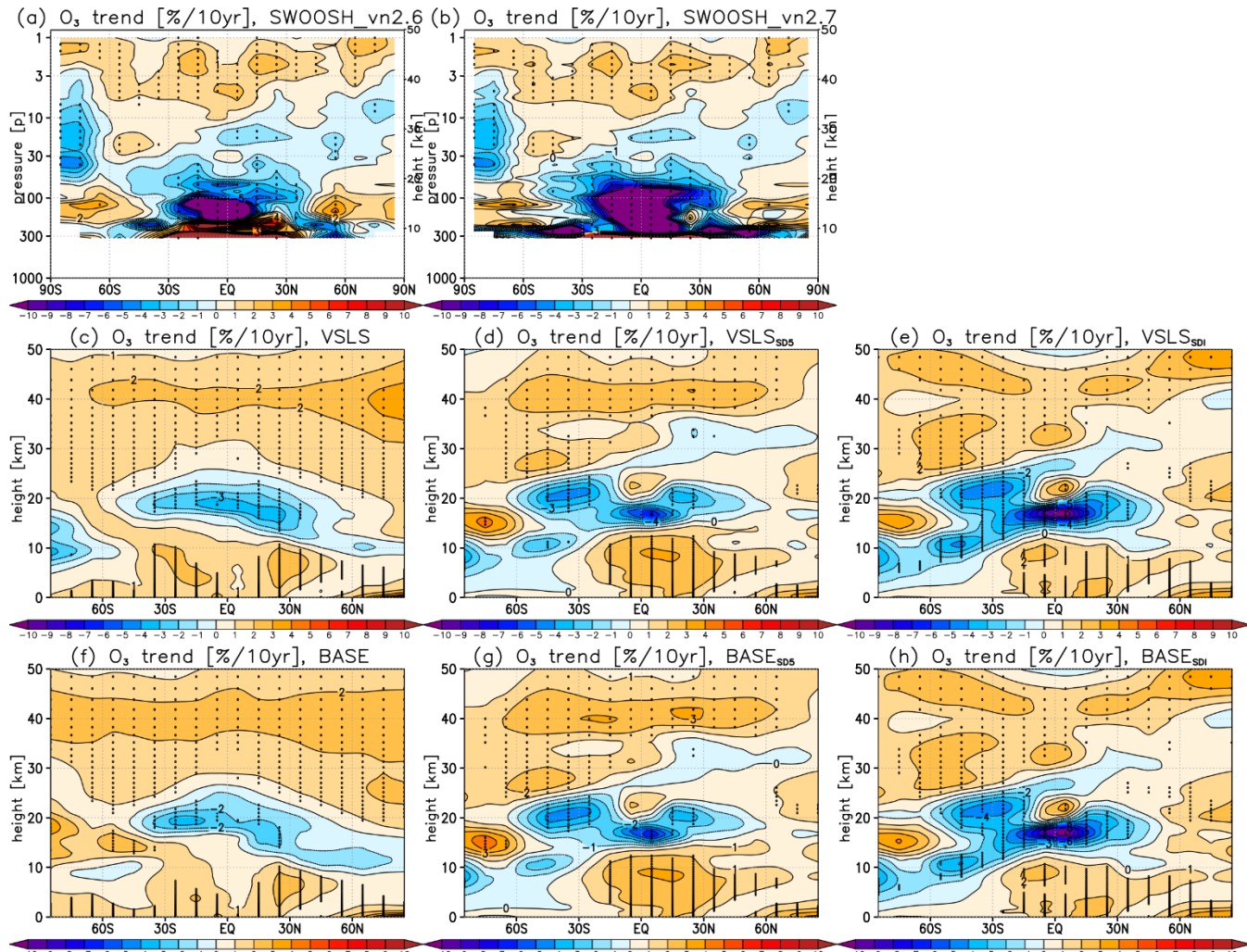

**Figure 4. The role of Cl-VSLS in contributing to the recent ozone trends.** Linear trends in de-seasonalised $O_3$ mixing ratios over December 1999 to August 2019 [% per 10 years] in (a-b) the SWOOSH vn2.6 and vn2.7 merged observational product and simulated in (c) the ensemble mean VSLS, (d-e) nudged $VSLS_{SD-5}$ and $VSLS_{SD-I}$, (f) ensemble mean BASE, and (g-h) the nudged $BASE_{SD-5}$ and $BASE_{SD}$-I. Hatching indicates statistical significance, here taken as regions where the magnitude of the derived trend exceeds ±2 standard errors.

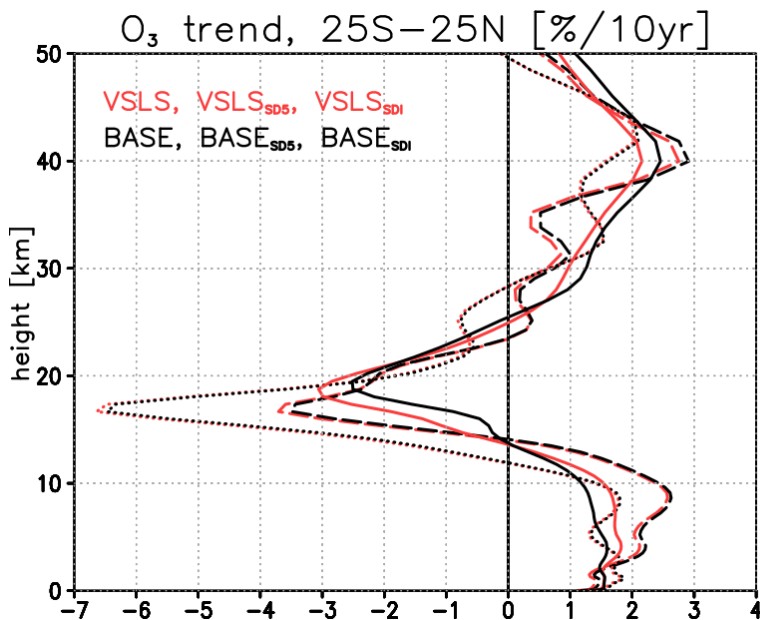

**Figure 5. Tropical ozone trends.** Linear trends in de-seasonalised $O_3$ mixing ratios over December 1999 to August 2019 [% per 10 years] averaged over the tropics. The simulations with Cl-VSLS included are in red, and the simulations without Cl-VSLS are in black. Solid lines are for the free-running simulations (VSLS and BASE), dashed lines are for the simulations nudged to ERA5 (VSLS$_{SD-5}$ and BASE$_{SD-5}$), and dotted lines are for the simulations nudged to ERA-Interim (VSLS$_{SD-I}$ and BASE$_{SD-I}$). See Fig. S4 in the Supplementary material for the corresponding changes in the mid-latitudes.
