# Peer review of "Atmospheric impacts of chlorinated very short-lived substances over the recent past - Part 2: Impacts on ozone"

_EGUsphere, 2023_

## Referee Comment (RC1)

**Review of the manuscript "Atmospheric impacts of chlorinated very short-lived substances over the recent past - Part 2: impacts on ozone" by Bednarz et al., EGUsphere preprint repository, 2023.**

The paper presents a modeling study using UM-UKCA with and without CL-VSLS halogens oriented to evaluate the impact of chlorinated very short-lived substances (VSLS) on recent-past stratospheric ozone trends (2010-2019). In addition, a new estimation of the ozone depletion potential (ODP) and stratospheric-ozone depletion potential (SODP) of the dominant CL-VSLS species (dichloromethane) is provided. The simulations are properly designed, the paper is well written and organized, and the analysis of the results is coherent. Their main conclusion is that CL-VSLS induce a modest but non-negligible role on the stratospheric ozone budget that counteract some of the gains achieved by the Montreal Protocol, but that the inclusion of CL-VSLS in UM-UKCS does not considerably modify the magnitude of diagnosed ozone trends in the tropical lower stratosphere. I suggest the paper is accepted for publication after the following issues have been solved:

**Main Comment:**

**Update the comparison with other modeling studies**

The authors properly compare their main results with previous publications performed by the group and/or co-authors, which are in line with the results presented here (i.e., recent ozone trends in the lower stratosphere are dominated by dynamics and not chemistry). However, a discussion with other results showing a significant contribution from VSLS chemistry (e.g., Villamayor et al., 2023) would improve the manuscript (see specific comments below). In addition, the discussion would benefit of a comparison of CL-VSLS impacts on ozone trends with respect to that arising from BR-VSLS (e.g., Sinnhuber and Meul, 2015; Barrera et al., 2020).

**Minor Comments:**

P1,L17-19: Are the values provided in the abstract "annual mean" or "springtime mean"? In case different time-averaging apply, please make it clear.

P2,L54-57: You should relate this with the recent publication by Villamayor et al. 2023 that found that inclusion of both natural and anthropogenic VSLs in a CCM results in a significant chemical signal that contribute to ozone reduction in the low-latitudes lower stratosphere.

P3,L74-75: It would be interesting to include some time of comparison between FR and SD. For example, a panel like Fig. 1c but presenting the mean +/- sigma of the latitudinal variation of ozone changes for the FR ensemble simulations. This would help to evaluate how well the FR range compares with the SD-runs values?. Note that Fig. 3 in part-1 of this paper (Bednarz et al., 2022) shows that FR simulations have a larger SGI, but similar SGI+PGI than SD simulations ... Thus I would expect a larger O3 influence for FR simulations. Is that the case?

P3,L80-81: What do you mean by "warm winters"? Are those 2-3 DU differences observed in Winter? or in Arctic spring preceded by warm winters? In addition, note the sentence only apply to high-latitudes, and is confusing as it mixes "warm winters" with "on average each year". Please rephrase and expand to make it clear.

P3,L81-81: Smaller over the Arctic, but larger over SH high-latitudes. Why is that? Due to the larger VSL-CL emissions in NH? Or could temperature changes play also a role here?.

P3,L86-87: Similar to P3,L74-75, you highlight the importance of nudging to evaluate the impact on Arctic results, but I wonder how large the difference between SD and FR simulations can be.

P4,L108-109: What is the rationale for showing ClO values at the beginning of march but O3 values at the end of march for nudged simulations? To evaluate the "cumulative" role of VSL-Cl chlorine on O3 losses?

P5,L135-136: As described, the chemical signal is negligible when SD are considered ... but induces an additional 0.5-1.0% enhancement for FR simulations (Fig S3). The authors may want to evaluate this and comment in more detail.

P5,L138-139: The results are in line with Chipperfield et al., 2018 and in contrast to Villamayor et al. 2023 ... which include both natural and anthropogenic VSLs. Note that Villamayor found also a negative trend, although not significant, when only VSLS-Cl were considered.

P5,L143-144: I suggest writing the numbers in the text as mean +/- sigma or mean +/- 2sigma ... as the range is already shown in Table 1.

P5,L151-152: I suggest mentioning explicitly that Cl-VSLS are not considered in the Montreal Protocol and/or amendments.

P5,L155-156: This absolute numbers for chlorine could be compared with VSL-Br impacts in the lower stratosphere as described by Sinnhuber and Meul, 2015 and Barrera et al., 2020.

P6,L156-157: Results summarized here apply mostly for the nudged simulations (SD), but not for the FR. A comment on this would be helpful.

P7,L192: I'm confused about the LBCs description: I thought for the case of CH2Cl2 and C2Cl4 an emission inventory (Claxton et al., 2020) was applied instead of using LBCs. Could you please make this clear?

P7,L203-204: Here you mentioned a source, but above you mentioned only LBCs were used. Please make it clear. In addition, if you provide a "perturbation value" here, it would be nice to provide the "base CH2Cl2 emission value" in previous paragraph, so one can easily estimate the magnitude of the perturbation. Similar to CFC-11, could you please provide the surface LBC for the base simulation.

**Language editing comments and Typos:**

P1,L16: Remove period "." after "time."

P4,L105: "shows that up to 25 ppt of the elevated ClO" or "shows differences up to 25 ppt for ClO". Please revise and make it clear.

P5,L158: What do you mean by "exact results"?

P7,L198: replace "ed" by "performed" or similar.

P8,L217: is it "where" or "when"

P8,L219: replace "included" by "including"

**Figures and Tables**

Figure 1: The final sentence of the caption should say annual global mean "total ozone column" changes ... similar for the "total ozone column change" magnitude that is missing in the Y axis of panel c.

Figure 2: Please use "ozone difference (%)" instead of the way it is written. Also I suggest including the dynamical tropopause line or 150 ppbv chemical tropopause line to help the reader splitting the troposphere and stratosphere.

Figure 3: Why do you show Ozone differences for 31 March but ClO differences for 1 March?

**References**

Barrera, J. A., Fernandez, R. P., Iglesias-Suarez, F., Cuevas, C. A., Lamarque, J.-F., and Saiz-Lopez, A.: Seasonal impact of biogenic very short-lived bromocarbons on lowermost stratospheric ozone between 60° N and 60° S during the 21st century, Atmos. Chem. Phys., 20, 8083–8102, https://doi.org/10.5194/acp-20-8083-2020, 2020.

Bednarz, E. M., Hossaini, R., Chipperfield, M. P., Abraham, N. L., and Braesicke, P.: Atmospheric impacts of chlorinated very short-lived substances over the recent past – Part 1: Stratospheric chlorine budget and the role of transport, Atmos. Chem. Phys., 22, 10657–10676, https://doi.org/10.5194/acp-22-10657-2022, 2022.

Chipperfield, M. P., Dhomse, S., Hossaini, R., Feng, W., Santee, M. L., Weber, M., Burrows, J.P., Wild, J.D., Loyola, D., Coldewey-Egbers, M.: On the cause of recent variations in lower stratospheric ozone. Geophysical Research Letters, 45, 5718– 5726. https://doi.org/10.1029/2018GL078071, 2018.

Claxton, T., Hossaini, R., Wilson, C., Montzka, S. A., Chipperfield, M. P., Wild, O., Bednarz, E. M., Carpenter, L. J., Andrews, S. J., Hackenberg, S. C., Mühle, J., Oram, D., Park, S., Park, M.-K., Atlas, E., Navarro, M., Schauffler, S., Sherry, D., Vollmer, M., Schuck, T., Engel, A., Krummel, P. B., Maione, M., Arduini, J., Saito, T., Yokouchi, Y., O'Doherty, S., Young, D., and Lunder, C.: A synthesis inversion to constrain global emissions of two very short lived chlorocarbons: dichloromethane, and perchloroethylene, J. Geophys. Res.-Atmos., 125, e2019JD031818, https://doi.org/10.1029/2019JD031818, 2020.

Sinnhuber, B.-M., and S. Meul (2015), Simulating the impact of emissions of brominated very short lived substances on past stratospheric ozone trends, Geophys. Res. Lett., 42, 2449–2456, doi:10.1002/2014GL062975.

Villamayor, J., Iglesias-Suarez, F., Cuevas, C.A. et al. Very short-lived halogens amplify ozone depletion trends in the tropical lower stratosphere. Nat. Clim. Chang. (2023). https://doi.org/10.1038/s41558-023-01671-y.

---

## Author Comment (AC1)

**AUTHORS RESPONSE TO REVIEWER #1**

The paper presents a modeling study using UM-UKCA with and without CL-VSLS halogens oriented to evaluate the impact of chlorinated very short-lived substances (VSLS) on recent-past stratospheric ozone trends (2010-2019). In addition, a new estimation of the ozone depletion potential (ODP) and stratospheric-ozone depletion potential (SODP) of the dominant CL-VSLS species (dichloromethane) is provided. The simulations are properly designed, the paper is well written and organized, and the analysis of the results is coherent. Their main conclusion is that CL-VSLS induce a modest but non-negligible role on the stratospheric ozone budget that counteract some of the gains achieved by the Montreal Protocol, but that the inclusion of CL-VSLS in UM-UKCS does not considerably modify the magnitude of diagnosed ozone trends in the tropical lower stratosphere. I suggest the paper is accepted for publication after the following issues have been solved:

We thank the reviewer for positive review and helpful comments that have improved the manuscript. We address the individual points below in blue.

**Main Comment:**

**Update the comparison with other modeling studies** The authors properly compare their main results with previous publications performed by the group and/or co-authors, which are in line with the results presented here (i.e., recent ozone trends in the lower stratosphere are dominated by dynamics and not chemistry). However, a discussion with other results showing a significant contribution from VSLS chemistry (e.g., Villamayor et al., 2023) would improve the manuscript (see specific comments below). In addition, the discussion would benefit of a comparison of CL-VSLS impacts on ozone trends with respect to that arising from BR-VSLS (e.g., Sinnhuber and Meul, 2015; Barrera et al., 2020).

We thank the reviewer for pointing this out – we have now modified the manuscript to include more discussion of the role of Cl-VSLS for ozone trends (Section 4), including comparison of trends inferred from nudged vs free-running simulations, as well as comparison with the recent study of Villmayor et al. (2023). We have also added some comparison with the impacts from Br-VSLS (as reported in the study of Barrera et al., 2020) into the discussion (Section 6).

**Minor Comments:**

P1,L17-19: Are the values provided in the abstract "annual mean" or "springtime mean"? In case different time-averaging apply, please make it clear.

We have now made this clear.

P2,L54-57: You should relate this with the recent publication by Villamayor et al. 2023 that found that inclusion of both natural and anthropogenic VSLs in a CCM results in a significant chemical signal that contribute to ozone reduction in the low-latitudes lower stratosphere.

We have now changed this part to read:

"While the effect of Cl-VSLS on the tropical lower stratospheric ozone trend in a chemistry-transport model has been estimated to be small (Chipperfield et al., 2018), a larger impact has recently been reported using a global chemistry-climate model containing a coupled troposphere-stratosphere chemistry scheme including chlorine, bromine and iodine VSLS (Villmayor et al., 2023), and as such the issue should still be re-examined."

P3,L74-75: It would be interesting to include some time of comparison between FR and SD. For example, a panel like Fig. 1c but presenting the mean +/- sigma of the latitudinal variation of ozone changes for the FR ensemble simulations. This would help to evaluate how well the FR range compares with the SD-runs values?. Note that Fig. 3 in part-1 of this paper (Bednarz et al., 2022) shows that FR simulations have a larger SGI, but similar SGI+PGI than SD simulations ... Thus I would expect a larger O3 influence for FR simulations. Is that the case?

Unfortunately, some of our output got corrupted, and as such we do not have the data to produce the corresponding changes in total column ozone in the free running VSLS and BASE simulations. We have, however, calculated the corresponding yearly mean changes in ozone concentrations analogous to Fig. 2a, and included it as Fig. S2 in the Supplementary Material. We have also added the following paragraph to Section 2:

"Given the significant dynamical variability characterising ozone levels on year-to-year timescales, we focus in this section on the results from the nudged model simulations. Whilst the corresponding free-running UM-UKCA simulations suggest higher Cl-VSLS induced lower stratospheric ozone losses (Fig. S2), consistent with the larger Cl-VSLS product gas to source gas stratospheric chlorine injection (Bednarz et al., 2022), there is large uncertainty in these values due to the contribution of natural variability."

P3,L80-81: What do you mean by "warm winters"? Are those 2-3 DU differences observed in Winter? or in Arctic spring preceded by warm winters? In addition, note the sentence only apply to high-latitudes, and is confusing as it mixes "warm winters" with "on average each year". Please rephrase and expand to make it clear.

We have now rephrased and clarified this part; we also now don't use the term 'warm winters' at all.

P3,L81-81: Smaller over the Arctic, but larger over SH high-latitudes. Why is that? Due to the larger VSL-CL emissions in NH? Or could temperature changes play also a role here?.

We apologize for the confusion. We have discovered a numerical problem with one of the simulations used (BASE$_{SD5}$) and have now re-run the simulation and updated the plots in the manuscript. We note that the correction does not substantially affect the conclusions of this paper, or the results of its accompanying PART 1 (Bednarz et al., 2022, https://doi.org/10.5194/acp-22-10657-2022). There is now, however, a much better agreement in the average Cl-VSLS induced ozone between the two sets of nudged simulations (Fig. 1c that the reviewer is referring to).

P3,L86-87: Similar to P3,L74-75, you highlight the importance of nudging to evaluate the impact on Arctic results, but I wonder how large the difference between SD and FR simulations can be.

See the response to P3, L74-75 above.

P4,L108-109: What is the rationale for showing ClO values at the beginning of march but O3 values at the end of march for nudged simulations? To evaluate the "cumulative" role of VSL-Cl chlorine on O3 losses?

We indeed chose to show ClO values at the beginning of March but O3 values at the end of March to evaluate the cumulative role of Cl-VSLS on ozone loss.. We also note that Arctic ClO values tend to peak in late winter/early spring, while zone losses tend to be largest later on in the season.

Such an approach has been previously used in many published studies of Arctic ozone depletion (e.g. Manney et al., 2011, doi:10.1038/nature10556; 2020, doi:10.1029/2020GL089063).

P5,L135-136: As described, the chemical signal is negligible when SD are considered ... but induces an additional 0.5-1.0% enhancement for FR simulations (Fig S3). The authors may want to evaluate this and comment in more detail.

We thank the reviewer for the useful suggestion. We have now expanded Section 4 to include more discussion of this; we have also added the middle panel of the old Fig. S3 that shows the tropical ozone trends in each simulation into the main manuscript (now Fig. 5).

P5,L138-139: The results are in line with Chipperfield et al., 2018 and in contrast to Villamayor et al. 2023 ... which include both natural and anthropogenic VSLs. Note that Villamayor found also a negative trend, although not significant, when only VSLS-Cl were considered.

As per response above, we now include more discussion (including references to the recent Villmayor et al. study) in the manuscript.

P5,L143-144: I suggest writing the numbers in the text as mean +/- sigma or mean +/- 2sigma ... as the range is already shown in Table 1.

Corrected.

P5,L151-152: I suggest mentioning explicitly that Cl-VSLS are not considered in the Montreal Protocol and/or amendments.

Thank you – we have now included this.

P5,L155-156: This absolute numbers for chlorine could be compared with VSL-Br impacts in the lower stratosphere as described by Sinnhuber and Meul, 2015 and Barrera et al., 2020.

We have now added the suggested comparison with the Br-VSLS impacts:

"In comparison, the ozone loss from the natural brominated VSLS emissions during the same time was estimated at ~1-2 DU in the tropics and ~5-6 DU in the midlatitudes (Barrera et al., 2020), albeit using a different climate model."

P6,L156-157: Results summarized here apply mostly for the nudged simulations (SD), but not for the FR. A comment on this would be helpful.

We now clarified this in the text.

P7,L192: I'm confused about the LBCs description: I thought for the case of CH2Cl2 and C2Cl4 an emission inventory (Claxton et al., 2020) was applied instead of using LBCs. Could you please make this clear?

Section M1 of the Methods is correct, i.e. LBCs of all four Cl-VSLS are used in these transient simulations as a source of Cl-VSLS (instead of emission inventory); this is the same approach as used by the PART1 of this study (Bednarz et al., 2022, https://doi.org/10.5194/acp-22-10657-2022). While we have performed simulations with $CH_2Cl_2$ and $Cl_2Cl_4$ emissions following the emission inventory of Claxton et al. (2020), these simulations are not used in either this manuscript or its companion PART1, but instead will form a future follow up PART3 study. We apologize for the confusion.

P7,L203-204: Here you mentioned a source, but above you mentioned only LBCs were used. Please make it clear. In addition, if you provide a "perturbation value" here, it would be nice to provide the "base CH2Cl2 emission value" in previous paragraph, so one can easily estimate the magnitude of the perturbation. Similar to CFC-11, could you please provide the surface LBC for the base simulation.

We apologize for the confusion – we now clarify all of this.

**Language editing comments and Typos:**

P1,L16: Remove period "." after "time."

Corrected.

P4,L105: "shows that up to 25 ppt of the elevated ClO" or "shows differences up to 25 ppt for ClO". Please revise and make it clear.

Corrected.

P5,L158: What do you mean by "exact results"?

We have clarified this as the results in particular regions and seasons.

P7,L198: replace "ed" by "performed" or similar.

Corrected.

P8,L217: is it "where" or "when"

Corrected.

P8,L219: replace "included" by "including"

Corrected.

**Figures and Tables**

Figure 1: The final sentence of the caption should say annual global mean "total ozone column" changes ... similar for the "total ozone column change" magnitude that is missing in the Y axis of panel c.

Corrected.

Figure 2: Please use "ozone difference (%)" instead of the way it is written. Also I suggest including the dynamical tropopause line or 150 ppbv chemical tropopause line to help the reader splitting the troposphere and stratosphere.

Corrected, and the tropopause added.

Figure 3: Why do you show Ozone differences for 31 March but ClO differences for 1 March?

As discussed above, Arctic ClO values tend to peak in late winter/early spring, while the ozone losses tend to be largest later on in the season (when the cumulative impact of halogen chemistry is largest). We note that such an approach has been previously used in many published studies of Arctic ozone depletion (e.g. Manney et al., 2011, doi:10.1038/nature10556; 2020, doi:10.1029/2020GL089063).

---

## Author Comment (AC2)

**AUTHORS RESPONSE TO REVIEWER #2**

**Scientific Significance:** The scientific questions addressed in this paper are appropriate for ACP. This work addresses for the first time the recent (2010-2019) impact of chlorinated Very Short-Lived Substances (VSLS) on ozone. The authors state they found "modest and non-negligible role of Cl-VSLS" to the stratospheric ozone budget. They also emphasized that continued Cl-VSLS emissions "could offset some gains by the Montreal Protocol". They were the second group to "estimated" the ODP of dichloromethane. I highly recommend this work for publication. Below are a few comments that may add to the impact of this work.

**Scientific Quality:** This work is of high scientific quality. The authors use a state-of-the-art CCM (UM-UKCA) nudged to ERA-Interim and ERA-5 meteorology. They also show results from an ensemble mean from the UM-UKCA CCM. The Cl-VSLS chemistry is also represented in a detailed manner.

**Presentation Quality:** Generally, in good shape. However, I would increase the font size of Figures 3 and 4.

We thank the reviewer for positive review and helpful comments that have improved the manuscript. We address the individual points below in blue.

**Specific Comment.**

Abstract, line 16. Typo "for the first time. Using the"

Corrected

Abstract, lines 19. I found it confusing in the abstract when the authors highlight the 2011, 2014, and 2020 years and state that up to 5-6DU monthly and zonal mean Arctic ozone reductions are simulated. Then in line 20 they highlight year 2020 with "~6DU ozone in total by the end of March". They state 2020 was a recent cold winter. I would suggest reworking sentences here being more specific why you picked 2011 and 2014 relative to 2020?

We apologize for the confusion. We have discovered a numerical problem with one of the simulations used (BASE$_{SD5}$) and have now re-run the simulation and updated the plots in the manuscript. We note that the correction does not substantially affect the conclusions of this paper, or the results of its accompanying PART 1 (Bednarz et al., 2022), but it does reduce the Cl-VSLS induced ozone loss inferred from the runs nudged to ERA5 in 2014. We have now changed the text accordingly.

Abstract, lines 21-23. The authors state that Cl-VSLS "do not considerably modify the magnitude of the recent ozone trends". Please be more specific, is this is tropical, polar, global, everywhere, etc? Also, why would one expect the trend to be significant over a short period (i.e., 2010-2019), especially in the Arctic? I would suggest adding more detail in the abstract if you want to mention ozone trend results.

We apologize for the confusion. While that particular statement in the abstract is fairly generic, i.e. relate to ozone trends in general in most of the stratosphere, the focus of our

study is to explain the persistent, statistically significant negative ozone trends diagnosed from the observations in the extra-polar lower stratosphere (Ball et al., 2018; 2019).

We have now clarified this in the abstract: "Despite ~doubling of Cl-VSLS contribution to stratospheric chlorine over the early 21st century, the inclusion of Cl-VSLS in the nudged simulations does not substantially modify the magnitude of the simulated recent ozone trends and, thus, do not help to explain the persistent negative ozone trends that have been observed in the extra-polar lower stratosphere."

Abstract, line 22. The ODP of Cl-VSLS is quantified. The paper mentioned this was the second ODP derivation. What do you mean that it was "estimated" – is that a typical way to discuss the derivation of an ODP? This topic deserves a couple sentences to clarify why you feel it is important to put this discussion in the abstract. See my comment on lines146-147 below.

We have now change 'estimated' to 'calculated', as well as added the following discussion to Section 5: "ODP is an important and well-established metric that is reported in WMO/UNEP Ozone Assessment Reports and other policy-facing documents to gauge the possible ozone depleting effect of a gas relative to CFC-11. Unlike for long-lived species, there are few explicit (i.e. based on global model calculations) ODP estimates of VSLS in the literature. This in part reflects the relative complexity of a VSLS ODP calculation, which requires consideration of both the source gas and product gas injection of halogens to the stratosphere. A sensitivity of the ODP to emission location and season can also play a role for some species (e.g. Brioude et al., 2010). Given the significant upward trend in the CH2Cl2 production and emission from its predominantly industrial source, the quantification of ODP for CH2Cl2 is particularly important."

Lines 46-49. "We showed that the contribution from these Cl-VSLS to stratospheric chlorine had increased from 70 ppt Cl in 2000 to 130 ppt Cl in 2019, i.e. almost doubling over the first two decades of the 21st century." One could make an argument that this information was taken from Bednarz et al., 2022, Part 1 – but it would be nice have this information brought to the abstract level when summarizing the trend results.

We agree and have now included this in the abstract.

Lines 53-54. In addition to Chipperfield et al. 2018, Wargan et al., 2018, and Orbe et al., 2020, Stone et al. also came to this conclusion (that dynamical variability is driving the O3 trend) using a chemistry climate model similar to UM-UKCA. Stone, K. A., Solomon, S., & Kinnison, D. E. (2018). On the identification of ozone recovery. Geophysical Research Letters, 45. https://doi.org/10.1029/2018GL077955.

We have added the reference to the Stone et al. study.

Line 85-86. "Furthermore, no significant Cl- VSLS-induced Arctic ozone loss can be diagnosed from the model ERA-Interim nudged monthly and zonal mean data for the spring 2011; this might be related to the small size of the polar vortex in that year and thus difficulties in reproducing its dynamical properties in a nudged model setup." This sentence is a bit disconcerting in that the reader is meant to figure why there are difference in the choice of reanalysis products. The main question in my mind is why even show ERA-Interim in this study? Presumably ERA-5 is the best ECMWF product to look at nudged ozone trends?

We agree with the reviewer. However, in our previous work (PART1, Bednarz et al., 2022) we explicitly discussed the role of the choice of reanalysis used for nudging for the simulated Cl-VSLS stratospheric input and the stratospheric chlorine budget. As such, we believe it is important to include the results made with both reanalysis datasets for completeness.

We have now expanded the discussion as to the possible reasons behind the differences in the simulated ozone responses: "We note that while very similar average large scale ozone losses are diagnosed from the simulations nudged to different reanalysis products (Fig. 1c), some differences can emerge for individual regions and seasons. In particular, no significant Cl-VSLS-induced Arctic ozone loss is diagnosed for the spring 2011 from the model nudged to ERA-Interim, while the Arctic ozone loss modelled in the spring of 2014 is notably higher in that run than in the run nudged to ERA5. This might be related to the generally small and variable size and structure of the NH polar vortex, thus difficulties in reproducing its dynamical properties in a nudged model set up, or to the differences in the resolved transport between the two reanalyses (e.g. Diallo et al., 2021; Ploeger et al., 2021; Bednarz et al., 2022). These results thus suggest that the choice of reanalysis for nudging could also be important in some years for the diagnosed ozone impacts from Cl-VSLS."

Figure 3 and 4. Please increase the font size of the titles and x-axis.

Done.

Lines 114-118. This is a very interesting discussion, i.e., "the impact of curbing emissions of long-lived ODSs achieved by the Montreal Protocol was estimated to reduce the magnitude of the Arctic ozone depletion in that spring by up to ~35 DU in mid-March compared to peak halogen levels in early 2000 (Feng et al., 2021)." It might be useful to bring this comparison of curbing the emissions of long-lived ODSs achieved by the Montreal Protocol up to the abstract level (i.e., versus 6 DU from Cl-VSLS)?

We agree that this is an interesting and relevant discussion. We note, however, that the result of Feng et al. was derived using a different climate model (i.e. TOMCAT/SLIMCAT chemistry-transport model), and as such a close abstract-level direct comparison may be misleading without sufficient amount of details. We now clarify the use of a different climate model in the text.

Lines 146-147. "The calculated stratospheric ODP of 0.0102 (confidence interval of 0.0062-0.0163) is similar to the whole atmosphere ODP metric, implying that $CH_2Cl_2$ has a relatively small effect on ozone below the tropopause in UM-UKCA." This is an interesting result. Is there anything more you can say about this result? Is this due to where $CH_2Cl_2$ is emitted (e.g., China)?

We have now added the following discussion of this result to Section 5: "In part, this reflects the relatively long tropospheric lifetime of $CH_2Cl_2$ (~100 days in the boundary layer; Hossaini et al., 2019), especially compared to some particularly short-lived iodine species (e.g. $CF_3I$) for which the distinction between ODP and SODP can be particularly important (Zhang et al. 2020)."